# Effects of Metformin Delivery via Biomaterials on Bone and Dental Tissue Engineering

**DOI:** 10.3390/ijms232415905

**Published:** 2022-12-14

**Authors:** Minjia Zhu, Zeqing Zhao, Hockin H. K. Xu, Zixiang Dai, Kan Yu, Le Xiao, Abraham Schneider, Michael D. Weir, Thomas W. Oates, Yuxing Bai, Ke Zhang

**Affiliations:** 1Department of Orthodontics, Beijing Stomatological Hospital, School of Stomatology, Capital Medical University, Beijing 100050, China; 2Biomaterials & Tissue Engineering Division, Department of Advanced Oral Sciences and Therapeutics, University Maryland School of Dentistry, Baltimore, MD 21201, USA; 3Center for Stem Cell Biology & Regenerative Medicine, University of Maryland School of Medicine, Baltimore, MD 21201, USA; 4Department of Stomatology, Beijing Friendship Hospital, Capital Medical University, Beijing 100050, China; 5Department of Oncology and Diagnostic Sciences, University of Maryland School of Dentistry, Baltimore, MD 21201, USA

**Keywords:** metformin, bone tissue engineering, osteogenesis, angiogenesis, inflammation regulation, bone defect, periodontitis

## Abstract

Bone tissue engineering is a promising approach that uses seed-cell-scaffold drug delivery systems to reconstruct bone defects caused by trauma, tumors, or other diseases (e.g., periodontitis). Metformin, a widely used medication for type II diabetes, has the ability to enhance osteogenesis and angiogenesis by promoting cell migration and differentiation. Metformin promotes osteogenic differentiation, mineralization, and bone defect regeneration via activation of the AMP-activated kinase (AMPK) signaling pathway. Bone tissue engineering depends highly on vascular networks for adequate oxygen and nutrition supply. Metformin also enhances vascular differentiation via the AMPK/mechanistic target of the rapamycin kinase (mTOR)/NLR family pyrin domain containing the 3 (NLRP3) inflammasome signaling axis. This is the first review article on the effects of metformin on stem cells and bone tissue engineering. In this paper, we review the cutting-edge research on the effects of metformin on bone tissue engineering. This includes metformin delivery via tissue engineering scaffolds, metformin-induced enhancement of various types of stem cells, and metformin-induced promotion of osteogenesis, angiogenesis, and its regulatory pathways. In addition, the dental, craniofacial, and orthopedic applications of metformin in bone repair and regeneration are also discussed.

## 1. Introduction

Metformin is an antidiabetic agent used as a first-choice medication for type II diabetes mellitus. With the chemical formula C4H11N5 (*N*,*N*-dimethyl biguanide), metformin is a small-molecule compound that has no known metabolites [1]. The effects and mechanisms of metformin have been extensively studied and reviewed; however, new insights and therapeutic uses of metformin are still being discovered. In vitro and in vivo studies have suggested that metformin also has the potential to protect the cardiovascular system, alleviate the effects of aging, ameliorate excess blood lipids, and promote osteogenesis and hard tissue regeneration [2]. A deeper understanding of the mechanism of the signaling pathways and the target tissues of metformin provided compelling evidence that metformin can be applied to treat various diseases, including coronavirus disease 2019 (COVID-19) [3]. Importantly, metformin has demonstrated a significant contribution to repairing bone defects and promoting angiogenesis [4,5,6,7,8].

Bone tissue engineering is a promising method that uses seed-cell-scaffold drug delivery systems to reconstruct bone defects caused by trauma and tumors, as well as other applications, such as dental hard tissue regeneration and periodontal regeneration [9]. The effects of metformin in bone tissue engineering and its promising clinical applications could contribute to diagnosis and treatment of bone defects (Figure 1). Metformin can be delivered locally via scaffolds to enhance osteogenesis and angiogenesis, and the underlying mechanism of its effects is believed to involve the activation of AMP-activated kinase (AMPK) or non-AMPK-specific pathways. Osteogenesis and adipogenesis are competing and reciprocal pathways; therefore, hyperglycemic conditions are more likely to promote the adipogenic differentiation of the target tissue [10]. Metformin has the ability to enhance osteogenesis by ameliorating hyperglycemic conditions, regulating sugar metabolism, and promoting cell migration [11]. In addition, metformin promotes bone mineralization and bone defect regeneration through different target tissues, including stem cells and non-stem cells. Emerging evidence shows that several types of stem cells could be driven toward the osteogenic lineage in the presence of metformin [12,13]. Previous studies also reported that metformin could facilitate the proliferation and osteogenesis of periodontal ligament stem cells (PDLSCs) and dental pulp stem cells (DPSCs) [9,14].

The key to effective bone and dental hard tissue regeneration is the proper coupling of osteogenesis and angiogenesis [15,16]. Large bone defects do not self-heal when left untreated [17]. Self-healing osteogenesis alone cannot repair large bone defects, and external interventions and regenerative techniques are required to solve this major challenge in bone tissue engineering. Promoting angiogenesis and vascular capillaries is a promising strategy for therapeutic bone regeneration [18]. In recent years, research has focused on the contradictory effects of metformin, and the mechanisms of the anti-angiogenesis and angiogenesis of metformin are under investigation [12,19,20].

Metformin also has the ability to inhibit chronic inflammation to promote osteogenesis indirectly. The anti-inflammatory effects of metformin were achieved not only by altering body metabolic parameters, such as hyperglycemia and insulin resistance, but also by activating anti-inflammatory pathways or the immune system. Metformin was reported to have a direct anti-inflammatory action by inhibition of nuclear factor κB via adenosine monophosphate activated protein kinase (AMPK) dependent and independent pathways [21,22]. AMPK is involved in various upstream and downstream pathways such as the AMPK/mechanistic target of rapamycin kinase (mTOR) pathway in mammals, extracellular regulated kinase (ERK)/AMPK with carbon dots, and the liver kinase B1 (LKB1)/AMPK pathway [23]. In addition, metformin has a dose-dependent effect. A recent study showed that a low dose of metformin could activate the presenilin enhancer protein (PEN2)-ATPase-AMPK pathway, whereas a higher dose acted via the AMP/LKB1/AMPK pathway [24]. In addition to AMPK pathways, metformin could also regulate bone tissue engineering via immune regulation or regulatory T lymphocytes, which is described in more detail later in this review. 

This is the first review of the effects of metformin on bone tissue engineering. This article reviews the recent cutting-edge research on the effects of metformin delivery via biomaterials on bone and dental tissue engineering. This includes metformin delivery via tissue engineering scaffolds, the effect of metformin concentration and dosage, metformin induced enhancements of various types of stem cells, and metformin-induced promotion of osteogenesis and angiogenesis. In addition, the dental, craniofacial, and orthopedic applications of metformin in bone and dental tissue repair and regeneration are also discussed.

## 2. Metformin Promotes Bone Repair and Regeneration

Bone defects are often caused by trauma, tumors, and other chronic diseases, such as periodontitis. Periodontitis is an inflammatory disease caused by plaque [25]. Traditional periodontal treatments have the ability to alleviate inflammation and suppress the progression of the inflammatory process to a certain extent [26]. However, when bone loss caused by periodontitis reaches a critical size, large bone defects cannot be repaired if treated only by the traditional treatments mentioned above. It remains an unsolved problem in restoring the structure and function of periodontitis bone loss. 

Our group developed drug delivery bone tissue engineering systems for regeneration applications. Among them, a calcium phosphate cement (CPCs)/alginate-hydrogel-microfiber (MF) scaffold system was developed to protect seed cells and transfer drugs to their target tissues. When seeded in this scaffold system, cells have relatively good biocompatibility and viability (Figure 2A–I) [9]. Moreover, this scaffold system also has an ideal mechanical strength and porous structure and could degrade within 3 to 4 days, releasing the encapsulated cells and drugs. CPCs have similar flexural strength, sheer strength, and other mechanical properties to alveolar bone. They also have a certain effect on osteo-induction [27]. MF could provide a porous structure for cells to seed, proliferate, and differentiate. When MF degrades, a microvascular-like structure could replace it, which has positive effects on angiogenesis [9]. Studies have shown that the incorporation of metformin into the CPC-MF scaffold system had no negative effect on cell viability compared with the scaffold system without metformin (Figure 2J–K) [9,28]. Moreover, the CPC-MF scaffold system is injectable and can form different shapes before its coagulation, which could better adapt to irregular bone defect sites, such as narrow, deep, and complex periodontal pockets. 

Zhao et al. [9] showed that when seeded into the CPC-MF scaffold, hPDLSCs proliferated, osteo-differentiated, and angio-differentiated well. Seed cells exhibited high expression levels of osteogenic genes, ALP (encoding alkaline phosphatase, RUNX2 (encoding RUNX family transcription factor 2), OCN (encoding osteocalcin), and OSX (encoding osterix) at day 14. The peak osteogenic gene expression levels of the ‘metformin + osteogenic’ group were significantly higher than those of osteogenesis only group (without adding metformin) at days 7 and 14. The gene expression values of groups with metformin were about three- to four-fold higher than those of the groups without metformin. In addition, the ALP activity of the groups with metformin were three-fold higher than that of the groups without metformin at day 14. Thus, metformin could enhance the ALP activity of hPDLSCs on the CPC-MF scaffold system. Cell-synthesized bone mineralization, stained with alizarin red (ARS), was deeper and denser with increasing culture time, indicating excellent bone regeneration capability and applicability in clinical practice. Groups with metformin showed darker ARS staining than all the groups without metformin. Quantitative analysis of bone matrix mineral synthesis by hPDLSCs on the CPC-MF scaffold system showed that the groups with metformin synthesized two- to three-fold more bone mineral than that of the groups without metformin at days 7 and day 14, respectively (Figure 2L–M).

The optimal concentration of metformin in bone repair and regeneration remains unknown. There are distinctive differences regarding its dosage either for different target tissues or different administration methods. Notably, an oral dose of 500–1500 mg metformin is widely used in clinical practice [29]. The following section reviewed both in vitro doses for local administration and in vivo doses of metformin.

Clinically relevant doses of metformin were demonstrated to be associated with the osteogenic differentiation and mineralization of iPSC-MSCs [30]. Notably, there are large differences in the optimal concentration of metformin for different cell types, suggesting that it is important to explore the most suitable concentration of metformin for its application in tissue engineering [30]. It was reported that metformin promotes osteoblastic differentiation through AMPK signaling at doses ranging from 0.5 to 500 μM [31]. This was specifically demonstrated via a dose-dependent effect on cell proliferation as well as an increase in extracellular mineral nodule formation and the most recognized osteogenic markers. Our previous study found that after treating MSCs cells with increasing doses of metformin (0–20 μM), metformin increased cell viability in a dose-dependent manner [31]. These findings underscore the importance of using therapeutically relevant doses when attempting to extrapolate in vitro results to the human clinical setting. Pharmacokinetic studies verified that within 2–4 h after an oral dose of 500–1500 mg, plasma concentrations of metformin in patients ranged from 2.7 to 20 μM [32]. 

Furthermore, sustained cell growth was observed under different treatment conditions. Sun et al. [7] reported that 125 μM was the optimal concentration of metformin for osteogenic differentiation and could promote implant osseointegration in rats. At concentrations over 200 µM, metformin inhibited the osteogenic differentiation of BMSCs [7]. The pro-osteogenic function of metformin during in vitro and in vivo osteogenesis of adipose-derived stromal cells was assessed [33]. The in vitro experiments showed that metformin added into the culture medium at a concentration of 500 µM promoted the differentiation of adipose-derived stromal cells into bone-forming cells and increased the formation of mineralized extracellular matrix. The in vivo models revealed that metformin at a dose of 250 mg/kg/day accelerated bone healing and facilitated new bone callus formation at fracture sites in a rat cranial defect model [31]. It was also reported that 100 µM metformin had a prominent positive effect on the osteogenic differentiation and a negative effect on the adipogenic differentiation of PDLSCs, which has important implications for the application of metformin in PDLSC-based osteogenesis and bone regeneration [34] (Table 1, Figure 3).
ijms-23-15905-t001_Table 1Table 1Effects of metformin concentration on different target stem cells.Target Stem CellsMetformin ConcentrationIn Vivo/In Vitro (Local Administration)The Effects of MetforminReferencesiPSC-MSCs0.5-500 μMIn vivoPromoted osteoblastic differentiation through AMPK signaling[30]0–20 μMIn vivoIncreased cell viability in a dose-dependent manner[31]BMSCs125 μMIn vivoThe optimal concentration[7]over 200 µMIn vivoInhibited the osteogenic differentiationAdipose-derived stromal cells500 µMIn vitroIncreased the formation of mineralized extracellular matrix[33]250 mg/kg/dayIn vivoAccelerated bone healing and facilitated new bone callus formationhPDLSCs100 µMIn vivoPositive effect on the osteogenic differentiationNegative effect on the adipogenic differentiation[34]Human endometrial stem cells10 wt%In vivoImprove osteogenic capabilityGBR application[13]In vitro10 to 15 wt%In vivoDecreased the viability of cell cultured on the surface In vitroDPSCs20 wt%In vitroRestored the tooth cavity, provided protection for dental pulp,[35]BMSCs0, 10, 50, 100, and 200 μMIn vivoIncrease mechanical strength and new bone volume[36]In vitroEffects of metformin concentration on different target stem cells. The optimal concentration of metformin, ranging from 0.5 to 500 μM, could have better osteogenic effects through the AMPK signaling pathway.
Figure 3Effects of metformin concentration on different target stem cells. The optimal concentration of metformin, ranging from 0.5 to 500 μM, could have better osteogenic effects through the AMPK signaling pathway. With different target stem cells, in vivo or in vitro (local administration), the dose-dependent effect may be different (red line). The effects of metformin, despite regulating more or less the same signaling pathways, differ depending on the target tissues [37], whether they are stem cells or non-cell materials. For seed cells, it mostly depends on their cellular origin and osteogenic lineage. In this section, we discuss the potential targeted seed cells from within or outside of the oral cavity. The oral cavity is a special environment for targeted stem cells. Its complexity derives from its composition [38]. For example, the periodontal complex consists of hard tissues (alveolar bones, cementum), soft tissues (periodontal ligament fibers, gingival tissues), blood vessels, and nerves [39]. Other than non-cell materials, stem cells are categorized into six sources: MSCs, dental-derived cells, BMSCs, hi-PSCs, cancer cells, and immune cells. In the following section, each of these sources is reviewed.
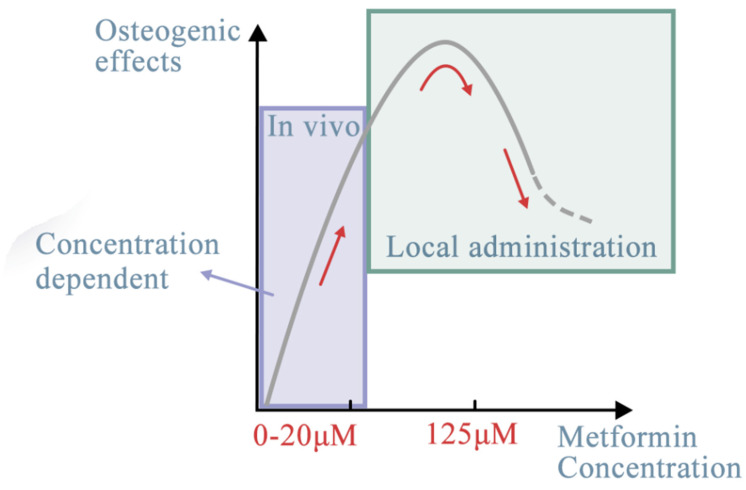


Local administration of metformin has also been studied. A novel guided bone regeneration (GBR) membrane containing polycaprolactone (PCL) and polyvinyl alcohol (PVA) with different concentrations of metformin was developed to improve osteogenic capability [13]. The results of that study confirmed the high potential of co-cultured stem cells with 10 wt% metformin PCL/PVA membranes for GBR applications [13]. Another model system containing 20 wt% metformin incorporated into resin was established for localized delivery of drugs into bone defect sites. The metformin-loaded resin could help restore the tooth cavity, provide protection for dental pulp, and prevent microleakage during restoration [35]. The effects of topical application of metformin on promoting new bone formation and remodeling were also evaluated [36]. Tendon–bone interface healing involves fine osteogenesis at the repair site. In vitro experiments used BMSCs with different concentrations of metformin (0, 10, 50, 100, and 200 μM) cultured together. The results showed that the mechanical strength and new bone volume of the metformin-applied group were significantly higher than those of the control group. However, the exact concentration and the pharmacokinetic curve of metformin should be defined in specific clinical applications. Further studies on the effect of metformin concentration are still needed.

A previous study showed that metformin could activate the AMPK pathways to induce the differentiation of MSCs into osteoblasts, promote their osteogenic differentiation, promote mineralization, and thus regenerate bone defects [39].

## 3. Metformin Enhances Angiogenesis

The hallmark of bone regeneration via bone tissue engineering is now considered as the reconstruction of vascular-like tissue [28]. In natural bone sites, alveolar bone is highly vascularized. In bone tissue engineering, newly formed regenerative bone tissue relies heavily on the supply of oxygen and nutrients provided by blood vessels. However, the speed of natural blood vessel formation is 5 μm per hour, which lags far behind the retraction rate because of their instability. Therefore, to reconstruct large bone defects, which have a larger diameter or complexity than critical sized bone defects, the pre-development of small capillary-like blood vessels is essential.

Despite its positive effect on osteogenesis, metformin is also considered to promote angiogenesis and pre-vascularization. In wound healing and the reformation of capillaries, metformin modulates the AMPK/mTOR/NLR family pyrin domain containing the 3 (NLRP3) inflammasome signaling axis [40] and has anti-inflammatory effects. During tissue ischemia, metformin also activates the AMPK/endothelial nitric oxide synthase (eNOS)–related pathway, promoting vascular remodeling [41]. A recent study showed that, when co-delivered with fibroblast growth factor 21 (FGF21), metformin promoted angiogenesis by upregulating Ang-1, recruiting endothelial cells. This provided a new strategy approach for diabetic wound management [42]. However, despite various studies showing that metformin has angiogenic effects, other studies report that metformin could inhibit the growth of cancer cells vis anti-angiogenic effects [43]. Angiogenesis is a key target for therapeutic intervention in various malignancies. Metformin could influence vascular cells and exert anti-angiogenesis-mediated effects by regulating microRNA (miRNA) signaling pathways. Studies show that metformin affects endothelial cell proliferation, migration, and related signaling pathways to participate in the regulation of angiogenesis [44]. There is no definite conclusion as to whether metformin has positive or negative effects on angiogenesis, and we discuss these effects in the following sections.

Furthermore, metformin can enhance angiogenesis in bone tissue engineering. A lack of vascular networks would lead to inadequate oxygen and nutrition supply in the large bone defect site. It would also fail to provide a suitable microenvironment for the proliferation and differentiation of seed cells, leading to failure to recruit other stem cells from autologous sources [12]. Tong et al. [45] developed a co-culture system for bone tissue engineering. In a mono-culture system, metformin showed positive angiogenesis exhibition effects on UC MSCs. Compared with the non-metformin-induced group, the metformin-treated group exhibited higher SCF and VEGFR gene expression. In the co-culture system, metformin could induce cell migration of HUVECs, helping MSCs to reconstruct vascular tissue and angiogenesis. In vivo, when HUVECs were co-cultured with stem cells from human exfoliated deciduous teeth (SHEDs), metformin could also enhance their angiogenesis effects [46]. In vivo, pre-treatment with metformin could induce cell-migration and angiogenesis and the expression of the angiogenesis-specific gene MAPK1. Therefore, using the CPC-MF scaffold system, we established the co-culture/tri-culture microenvironment system [28]. The co-culture system includes hPDLSCs, induced pluripotent stem cell (iPSC)-MSCs or other MSCs, human umbilical vein endothelial cells (HUVECs), and pericytes (Figure 4A–E). When different ratios and types of cells are cultured together, they could form capillary-like structures that could be identified using immunostaining (Figure 4C). In addition, much higher vessel length and numbers of junctions were found in the co-culture groups as well as higher levels of the angiogenic markers, vascular endothelial growth factor (VEGF), and osteogenic markers. When the co-culture microenvironment system was induced using metformin, newly formed blood vessels could be found at the replanting bone defect sites, indicating metformin’s ability to enhance angiogenesis [12].

## 4. Effect of Metformin on Dental-Derived Stem Cells

PDLSCs originate from the periodontal ligament and are the most well-known stem cells in bone tissue engineering. PDLSCs are also known for their triple differentiation potential: PDLSCs can differentiate into not only an osteogenic lineage but also into fibrogenic and cementogenic lineages to regenerate the periodontal complex [39] (Figure 5).

In bone tissue engineering, when seeding on different scaffolds, such as CPC or polydopamine-templated hydroxyapatite (tHA), PDLSCs could interact with metformin and exert different effects. CPC scaffolds have nearly the same mechanical characters as cancellous bone and can upregulate PDLSC osteogenic expression [9,47]. The scaffold tHA could induce apoptosis by producing reactive oxygen species (ROS). However, when combined with metformin, tHA notably decreases ROS production, enhances the level of microtubule-associated protein 1 light-chain 3 II and BECLIN-1 and eliminates cell injury and apoptosis, resulting in increased seed cell osteogenic gene expression. Similarly, metformin activates AMPK signaling pathways in the scaffolds despite their different upstream or downstream regulations. Moreover, PDLSCs could differentiate into fibrogenic lineages. Studies showed that metformin could regulate the fibrogenic cytokine transforming growth factor beta (TGF-β), representing a well-designed cascade treatment for cancer [48]. In conclusion, as an important family of target stem cells, hPDLSCs, have excellent differentiation potential in bone tissue regeneration.

Dental pulp cells (DPCs) could be harvested from dental pulp. As an important type of MSC, DPCs not only share similar gene expression with MSCs but also have multipotent differentiation potential. DPCs express functional organic cation transporter-1 (OCT-1), similar to other MSCs, for metformin intracellular uptake [49]. OCT-1 belongs to the solute carrier (SLC) superfamily and acts as a determinative uptake pathway of metformin pharmacokinetics [50]. Moreover, DPCs could reconstruct dentin-pulp-like complexes, differentiate into odontoblasts, and secret dentin matrix protein (DMP), type I collagen, and dentin sialo phosphoprotein (DSPP). The odontogenic differentiation of DPCs could be triggered by metformin through the classical AMPK signaling pathway [51]. Combined with scaffold materials, such as CPC, metformin could stimulate osteogenic and odontogenic differentiation of DPCs via controlled release and avoid rapid dilution [30] (Figure 6). 

Adipose-derived stem cells (ASDSCs) could originate from various sources, such as the oral cavity, epididymal fat pads (Epi-ADSCs), and inguinal fat pads (Ing-ADSCs). Metformin could activate ADSC AMPK signaling pathways, thus leading to inhibition of Cidec, perilipin1, and Rab8a [52]. Especially in a high glucose microenvironment, ADSC osteogenic differentiation and the levels of cell autophagy markers BECLIN1 and microtubule-associated protein 1 light-chain 3 (LC31/II) decreased. Metformin could protect ADSC osteogenic differentiation and reverse the negative regulation of a high glucose environment [53].

Dentin is a primary hard tissue, and dental pulp is a soft tissue, which together comprise the teeth. In deep caries, reparative dentin could be formed by odontoblasts. DPSCs are an effective pre-odontogenic lineage. For potential clinical use, pulp capping materials, resin, or scaffolds carrying metformin could induce DPSC odontogenic differentiation and mineralization [35,51,54,55].

In bone tissue engineering, SHEDs are ideal seed cells because of their relatively low immunogenicity, various differentiation potentials (osteoblasts, odontoblasts, chondrocytes, adipocytes, and nerve cells), higher proliferation rate, osteogenic and angiogenic potentials, and their double telomere length compared with other dental-derived stem cells [56]. Metformin could be introduced into scaffolds containing SHEDs to enhance their osteogenic differentiation. Moreover, metformin could be loaded onto mesoporous silica nanosphere (MSNs)-laden gelatin methacryloyl (GelMA) photocrosslinkable hydrogels [57]. This special hybrid material has both synergistic properties for metformin’s osteogenic effect on SHEDs and could be functionalized as an injectable filling for bone defect sites.

At a healthy bone site, osteoclasts and osteoblasts are in a dynamic equilibrium, and they form and degrade alveolar bone to reconstruct the bone shape. However, in diabetes, periodontitis, or other inflammatory conditions, the equilibrium breaks down, inducing osteoclast precursors, which upregulate bone degradation. At the cellular level, metformin facilitates osteoblasts and inhibits osteoclast formation and activity, when orchestrated by periodontal ligament fibroblasts [58]. That study indicated that metformin has the potential to reconstruct alveolar bones in bone defects caused by periodontitis.

## 5. Effect of Metformin in Other Cells and Biomaterials

MSCs are classical stem cells that could be isolated from various sources, such as BMSCs, adipose-derived stem cells (ADSCs), and umbilical cord mesenchymal stomal cells (UC MSCs). Among these, UC-MScs are inexpensive, easily harvested from extraembryonic sources, and have remarkable bone regenerative efficiency. Compelling evidence suggests that MSCs could ingest metformin by expressing cell membrane organic cation transporters (OCTs). UC-MSCs could activate the AMPK pathway by OCT-dependent cellular uptake of metformin. UC-MSCs-induced mineralizing nodule formation could also be facilitated by metformin. When an *OCT1* small interfering RNA (siRNA) duplex was transfected into UC-MSCs to block *OCT1* expression, metformin accumulation and Runx-2 expression decreased by approximately 40% (Figure 7) [31]. Furthermore, MSC-derived nanoparticles and exosomes could transport proteins, miRNA, and mRNA and exert the same features as MSCs [59]. Hence, further study needs to investigate the combination of exosomes and metformin-loaded scaffolds.

BMSCs are the gold standard cell type in bone tissue engineering and are a perfect control group in parallel with other mesenchymal stem cells [60]. BMSCs are acquired by harvesting alveolar bone particles from the implant bed during orthognathic surgery or from maxilla mandible bone marrow [7]. Several studies showed that the immune system could be a crucial player in bone reconstructing, and metformin reshapes bone metabolism by influencing macrophages (promoting the M2 phenotype and reducing the M1 phenotype) and the immune milieu. Metformin activates the phosphatidylinositol-4,5-bisphosphate 3-kinase (PI3K)/protein kinase B (AKT)/mTOR pathway to stimulate M2 macrophages, which could promote osteogenesis and improve immune–bone hemostasis [6].

iPSCs are upstream stem cells that can be differentiated into mesenchymal stem cells in vitro. In bone tissue engineering, iPSC-derived MSCs are considered to exhibit higher pluripotency and proliferative capabilities than the gold standard BMSCs. Pharmacologically, hi-PSCs, like other classical MSCs, express OCT-1, which allows the uptake of metformin, thus enhancing osteogenic differentiation and mineralization via the LKB1/AMPK signaling pathway [30]. Moreover, hi-PSC-derived liver and islet organoids were found to have a lower transportation capacity under a high glucose environment. However, when treated with metformin, hi-PSCs could alleviate both mitochondrial dysfunction and metabolically-relevant signaling pathways, thus recapitulating the human-relevant liver–islet axis [61].

Harvested from human umbilical cords, both hUVECs and UC-MSCs are highly potentialized clinical materials in regenerative medicine applications. hUVECs are indispensable endothelial cells, playing a key role in capillary-like tissue formation. In bone tissue engineering, metformin could enhance both osteogenesis and angiogenesis of UC-MSCs [45]. Moreover, OCT-1 expression in UC-MSCs allowed the uptake of metformin, thus exerting an angiogenic effect on UC-MSCs [31]. Studies showed that when loaded on PLA/PCL scaffolds, gelatin nanocarrier/metformin could be delivered into calvaria bone defects in a rat model, resulting in higher expression of the endothelial cell marker CD31 than that in the control group [12].

Other non-cell materials, such as scaffolds, macromolecule polymers, graphene, and resins could also carry and deliver metformin in bone tissue engineering. Studies revealed that graphene, a promising smart biomaterial, had both antibacterial effects and spectacular physical peculiarities [62]. In addition, black phosphorus (BP), an allotrope of phosphorus, showed the same properties but had better biocompatibility, biodegradability, and biosafety [63]. Furthermore, borophene could also act as a prototype for metformin-loaded scaffold materials [64]. Studies showed that polyurethane (PU) combined with metformin as a chain extender could develop a novel kind of biocompatible bone regenerative film. This new material could prevent soft tissue cells entering bone cavities and promote bone development and regeneration [65].

Metformin can target various tissues; however, the mechanisms remain unclear. First, although various studies have confirmed that metformin activates the classical AMPK signaling pathway, the upstream and downstream pathways remain unknown. Second, different doses of metformin could activate different pathways and have other unknown effects in bone tissue engineering. Third, there is controversy as to whether metformin has positive or negative effects on angiogenesis. In conclusion, the effects of metformin in bone tissue engineering should receive more research attention in the future.

## 6. Clinical Application of Metformin in Bone and Dental Tissue Regeneration and Development

Metformin-loaded scaffolds have broad application prospects in patients with periodontitis, tooth cavities, and other cranio-maxillo-facial defects [9]. Metformin has positive effects on bone metabolism and regeneration. The potential clinical application of metformin-loaded materials is reviewed in the following section.

Both in vivo and in vitro studies confirmed that when treated with metformin-loaded materials, treatment outcomes of patients improved significantly compared with those who did not receive metformin. Periodontal intrabony defects (PIDs) fall into the category of bone defects resulting from periodontitis. Various treatments have been developed for this clinical situation. The combination of metformin with scaffolds as bone grafts led to more improvement compared with simply performing open flap debridement (OFD) [66]. Bone defects caused by tumors have higher recurrence rates due to the difficulty to repair and eliminate residual tumor cells. Metformin-loaded poly (L-lactic acid) (PLLA)/nanoscale hydroxyapatite (nHA)/metformin (MET) nanocomposite scaffolds could simultaneously enhance bone repair and suppress bone tumors [67].

Furthermore, other than dental tissue regeneration, metformin also has outstanding effects in promoting bone regeneration in critical size bone defect models. As assessed using histological staining, a metformin-loaded scaffold gelatin/nano-hydroxyapatite/metformin scaffold (GHMS) appeared to preserve a higher alveolar ridge, form a lower percentage of connective tissue, and form more bone tissue compared with simple grafting techniques [68].

However, the concentration-dependent osteogenic effects of metformin are still unclear. Further studies are needed to use animal models to investigate how metformin interacts with various types of target tissues and to determine the osteogenic and angiogenic effects of metformin.

## 7. Conclusions

This is the first review on the effects of metformin on bone tissue engineering. Metformin-loaded biomaterials have high potential in a wide range of bone and dental tissue regeneration and development applications. The present article mainly reviewed the osteogenesis and angiogenesis effects of metformin and its concentration-dependent properties both in vivo and in vitro. In the range of 2–20 μM, MSC viability increases in a dose-dependent manner in vivo. In vitro, this particular concentration is similar to plasma concentrations of metformin in patients after 2–4 h of an oral dose of 500–1500 mg, ranging from 2.7 to 20 μM. The concentration after local administration is slightly higher, ranging from 125 to 200 μM, and could promote osteogenesis, thus being useful in bone tissue regeneration. Metformin promotes osteogenesis and angiogenesis through various different target tissues, including hPDLSCs. hPDLSCs are considered a classical stem cell component in bone tissue engineering. For example, scaffolds loaded with metformin seeded with hPDLSCs could generate greater amounts of new bone and pre-vascularization in forming capillary-like vessels and increased osteogenic gene expression, ALP, and mineral synthesis by three- to four-fold compared with those of the non-metformin-loaded control scaffold. Other dental-derived stem cells, such as DPCs, ADSCs, and SHEDs, could proliferate and differentiate into the osteogenic lineage under stimulation by metformin. Other cells with multi-pluripotency and scaffold biomaterials were also demonstrated to be potent alternative target tissues for the regeneration of bone defects. Therefore, metformin, a common drug originally designed for a specific disease, has several unique advantages in bone tissue engineering, representing a promising and exciting drug with utility beyond that for which it was first designed.

## Figures and Tables

**Figure 1 ijms-23-15905-f001:**
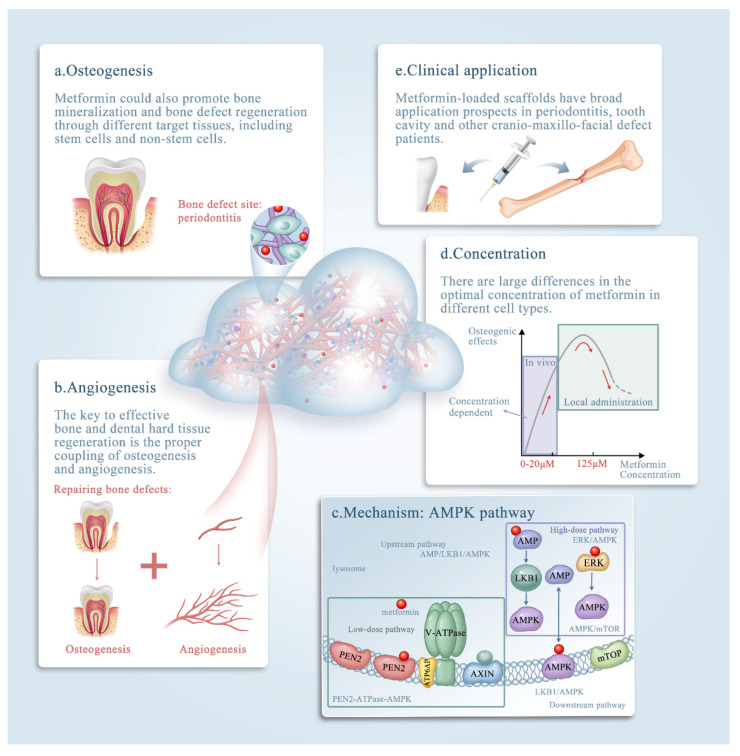
The roles of metformin in bone tissue engineering. The effects of metformin in bone tissue engineering. The promising clinical application metformin could contribute to diagnosis and treatment of bone defects. (**a**–**e**) Schematic illustrating metformin delivery via biomaterials, mainly via scaffolds, and its effects on bone and dental tissue engineering. Note that metformin exerts its effects by: (**a**) enhancing osteogenesis; (**b**) enhancing angiogenesis; (**c**) affecting the AMPK pathway; and (**d**) acting in a concentration-dependent manner. (**e**) The clinical applications of metformin. The underlying mechanism of metformin (through the AMP-activated kinase (AMPK) signaling pathway) is discussed in detail in the text.

**Figure 2 ijms-23-15905-f002:**
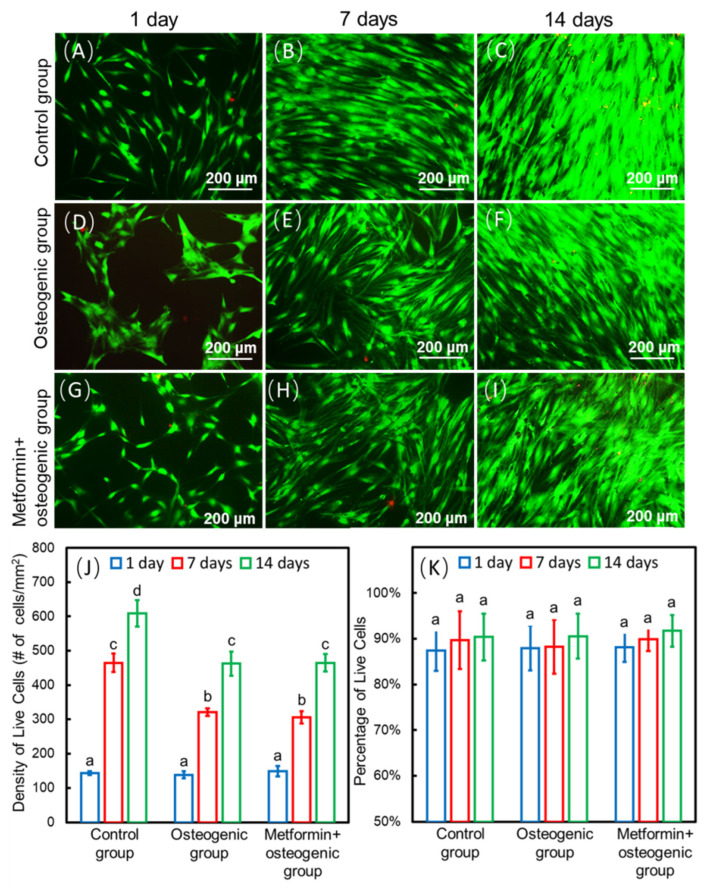
Methods of seed-cell drug delivery via a calcium phosphate cement (CPCs)/alginate-hydrogel-microfiber (MF) scaffold system. Live/dead staining images of cells seeding on the CPC-MF scaffold system at days 1, 7, and 14 (**A**–**I**); live cell density and percentages of live cells encapsulated in CPC-MF scaffold system (**J**,**K**); synthesis of bone minerals by the encapsulated stem cells, which was enhanced by the encapsulated metformin. Alizarin red (ARS) staining and cell-synthesized mineral quantification of human periodontal ligament cells (hPDLSCs) on CPC-MF scaffold system with or without metformin (**L**,**M**) Values with dissimilar letters (a,b,c) are significantly different from each other (*p* < 0.05). (Adapted from reference [9], with permission).

**Figure 4 ijms-23-15905-f004:**
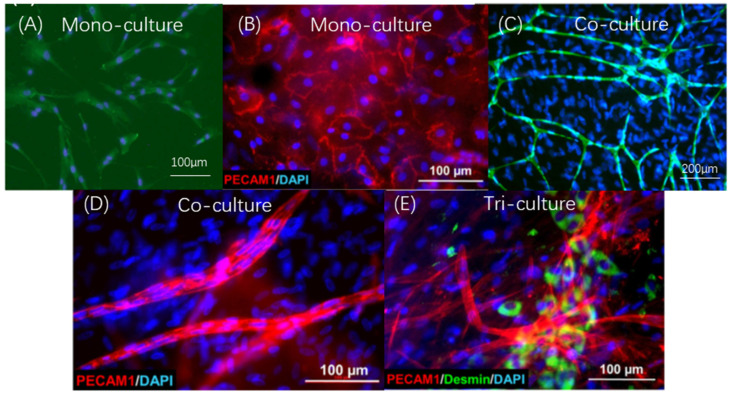
CPC-MF scaffold system and co-culture/tri-culture microenvironment system construction for bone regeneration. (**A**) hPDLSCs (second passage) were immunofluorescently stained with anti-STRO-1 antibodies (green) and 4′,6-diamidino-2-phenylindole (DAPI) for nuclei (blue). (**B**) Mono-cultured human umbilical vein endothelial cells (HUVECs) were immunofluorescently stained with endothelial marker platelet and endothelial cell adhesion molecule 1 (PECAM1; red) and DAPI for nuclei (blue) at day 15. (**C**) For the group co-cultured with HUVECs and hPDLSCs, vessel length/junctions were quantified. HUVECs were immunofluorescently stained with CD31 (green), and both HUVECs and hPDLSCs were stained with DAPI for nuclei (blue) at day 21. (**D**) Group co-cultured with hi-PMSCs and HUVECs. HUVECs were stained in red, and both nuclei were stained in blue. (**E**) Tri-culture group with hi-PMSCs, HUVECs, and pericytes. HUVECs were stained in red, pericytes were stained for the specific marker desmin in green, and all nuclei were stained in blue. (Adapted from references [9,28,37], with permission).

**Figure 5 ijms-23-15905-f005:**
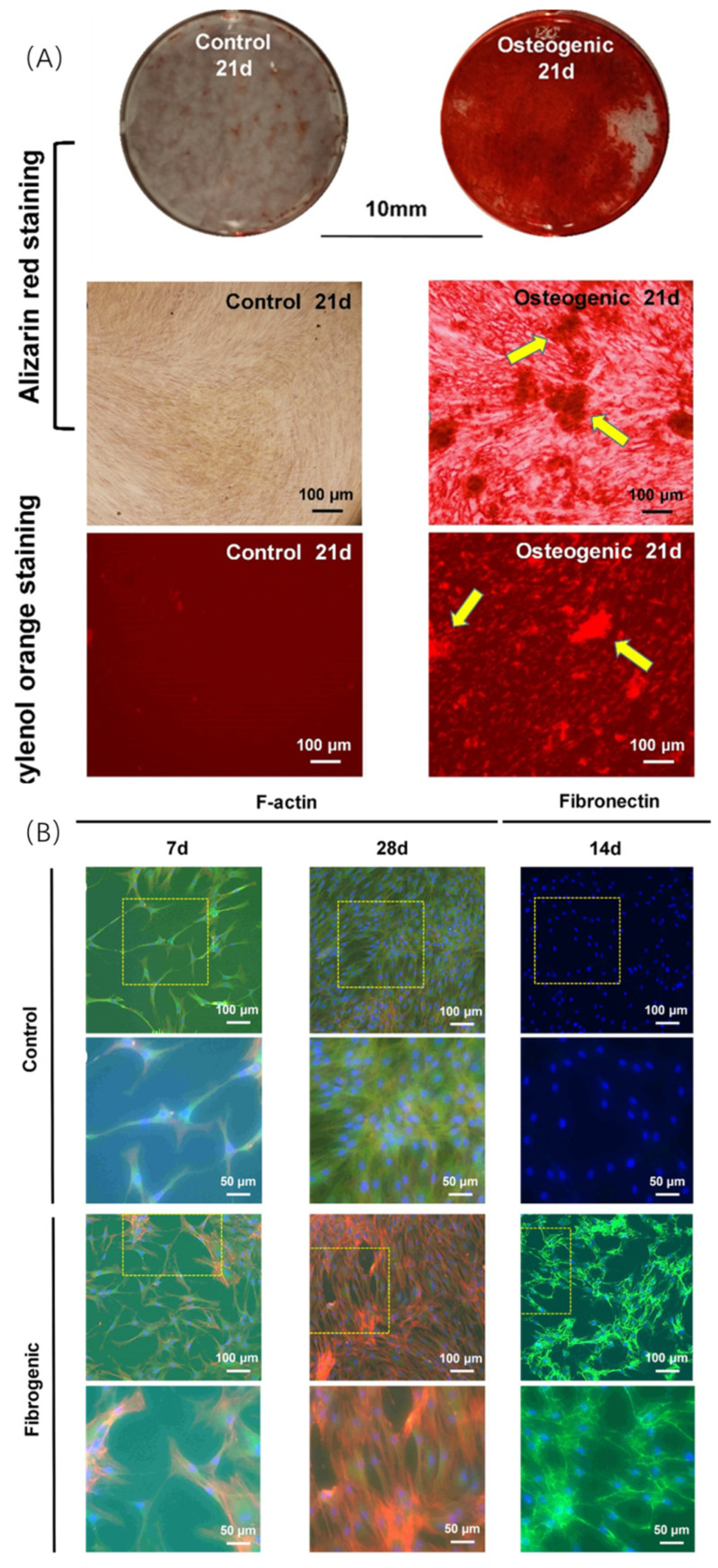
Triple differentiation potential of PDLSCs. (**A**) Osteogenic lineage. (**B**) Fibrogenic lineage. (**C**,**D**) Cementogenic lineage. Calf alkaline phosphatase (CAP) Values with dissimilar letters (a,b,c) are significantly different from each other (*p* < 0.05). (Adapted from reference [39], with permission).

**Figure 6 ijms-23-15905-f006:**
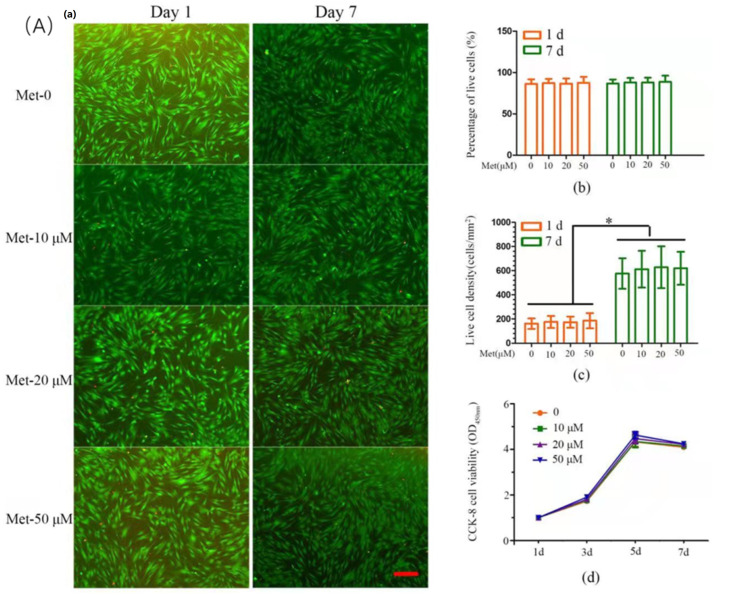
The effects of metformin on dental pulp cells (DPCs). (**A**) Effect of metformin on cell viability and proliferation of DPCs. (**a**) Representative live/dead images of metformin-treated DPCs at days 1 and 7 of culture, with live cells stained green and dead cells shown in red. In all four groups, live cells were abundant, and dead cells were few (scale bar = 100 μm). (**b**) Percentage of live cells of DPCs was around 90%. Data represent mean ± SD of 3 experiments with triplicates. (**c**) All groups exhibited increasing live cell density. Data represent mean ± SD of 3 experiments with triplicates. * *p* < 0.05. (d) Metformin has no effect on the cell proliferation. Data represent mean ± SD of 3 experiments with triplicates. (**B**) Effect of metformin-induced ALP activity and mineralized nodule formation in DPCs. (**a**,**b**) DPCs were treated with metformin (50 μM) in the absence or presence of Compound C (10 μM, pretreatment for 1 h); cells were retreated every 3 days. ALP activity (**A**) and ALP mRNA expression (**b**) were measured at each time point. Data represent mean ± SD of 3 experiments with triplicates. * *p* < 0.05. ** *p* < 0.001. (**c**) DPCs were cultured in osteogenic induction medium for 14 days, mineralized nodule formation was assessed by von Kossa staining (scale bar = 100 μm). (**d**) On the 14th day, the calcium content was determined. Data represent mean ± SD of 3 experiments with triplicates. * *p* < 0.05. ** *p* < 0.001. (Adapted from reference [51], with permission).

**Figure 7 ijms-23-15905-f007:**
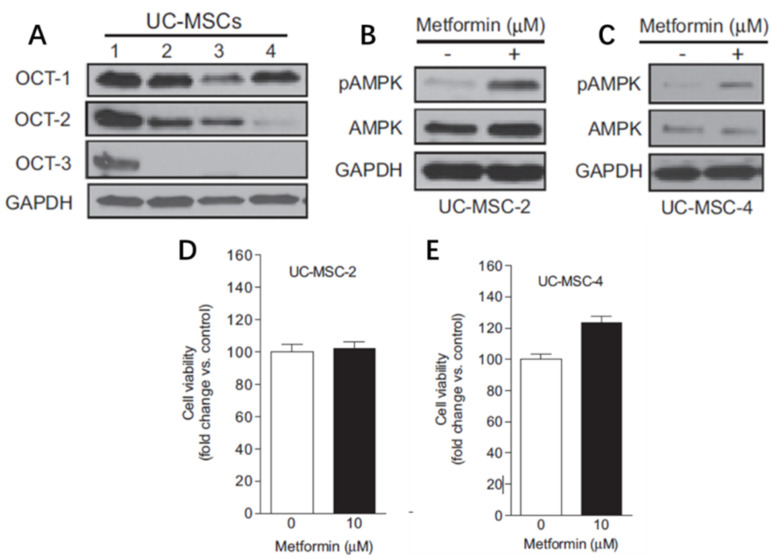
UC-MSCs expressing OCT to ingest metformin. (**A**–**C**) Organic cation transporter (OCT) protein levels in umbilical cord mesenchymal stomal cells (UC-MSCs). (**D**,**E**) Transfection of *OCT1* small interfering RNA (siRNA) duplex to downregulate metformin accumulation and RUNX family transcription factor 2 (Runx2) expression in UC-MSCs. AMPK, AMP-activated kinase; pAMPK, phosphorylated AMPK; GAPDH, glyceraldehyde-3-phosphate dehydrogenase (Adapted from reference [31], with permission).

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
