# Peer review of "Effects of Metformin Delivery via Biomaterials on Bone and Dental Tissue Engineering"

_ijms, 2022, doi:10.3390/ijms232415905_

Round 1

Reviewer 1 Report

This manuscript focuses on the effects of metformin on osteogenesis and angiogenesis, two intriguing parts of bone and dental regenerative application. It deals with an important research topic, and is well presented. However, some issues need to be revised, and are listed below:

1)      The word ‘cells’ from adipose-derived stem cells were not usually capitalized. This need to be changed on Page 15 and 16.

2)      The caption of Figure 3 is not clear. Please be more specific about whether it is mono-cultured or co-cultured.

3)      Please add more convincing studies regarding the effect of metformin on angiogenesis in Section 3.

4)      Figure 4 is not clear and is inconsistent with the paragraph above, it did not show whether metformin is applied in vivo or in vitro. Please be more specific.

5)      Since in Section 6, author indicates that many MSCs have OCT-dependent uptake of metformin. The authors would do well to discuss further proven studies involved OCT and MSCs (section 6 and 7).

6)      The subtitle of the manuscript on Section 7 is confusing. I suggest the author change them to ‘Effect of metformin in other cells and biomaterials’.

7)      There are more foreseeable future applications of metformin neglected in this review: quantifying metformin both in vivo and in vitro, how to apply metformin to its target, and its release curve, etc. This should be further addressed in the ‘clinical application’ section.

Once the above concerns are fully addressed, this review could be accepted for publication in this journal.

Author Response

December 03, 2022

Prof. Claire Lim

Editor, International Journal of Molecular Sciences

Dear Prof. Lim:

I hope you are doing very well.  Thank you very much for a favorable review on our paper, ijms-2044947, entitled: “Effects of metformin delivery via biomaterials on bone and dental tissue engineering.” The reviewers provided excellent comments, all of which have been addressed in the revised paper.  Below is a point-by-point response to the review comments.  Each comment is followed by our response in a red color.  The revised sections in the paper are also highlighted in red.

Reviewer 1

This manuscript focuses on the effects of metformin on osteogenesis and angiogenesis, two intriguing parts of bone and dental regenerative application. It deals with an important research topic, and is well presented. However, some issues need to be revised, and are listed below:

Thank you very much for your excellent review and comments that have helped significantly improve our paper.

1) The word ‘cells’ from adipose-derived stem cells were not usually capitalized. This need to be changed on Page 15 and 16.

Good Point. We have revised “Adipose-derived stem Cells (ASDSCs)” in page 15 and 16 to “Adipose-derived stem cells (ASDSCs)”

2) The caption of Figure 3 is not clear. Please be more specific about whether it is mono-cultured or co-cultured.

We agree. We have added subtitles onto Figure 3.

3) Please add more convincing studies regarding the effect of metformin on angiogenesis in Section 3.

Good point. We have added the following parts in Section 3: “A recent study showed that, when co-delivered with fibroblast growth factor 21 (FGF21), metformin promoted angiogenesis by upregulating Ang-1, recruiting endothelial cells. This provided a new strategy approach for diabetic wound management [35].”

4) Figure 4 is not clear and is inconsistent with the paragraph above, it did not show whether metformin is applied in vivo or in vitro. Please be more specific.

We are sorry for this misunderstanding. We have added “With different target stem cells, in vivo or in vitro (local administration), the dose-dependent effect may be different (red line).” in the following figure caption.

5) Since in Section 6, author indicates that many MSCs have OCT-dependent uptake of metformin. The authors would do well to discuss further proven studies involved OCT and MSCs (section 6 and 7)

Yes, good points. We have added the following parts in Section 5 (the original Section 6) line 376:“DPCs not only share similar gene expression with MSCs, but also have multipotent differentiation potential. DPCs express functional organic cation transporter-1 (OCT-1), similar to other MSCs, for metformin intracellular uptake [49]. OCT-1 belongs to the solute carrier (SLC) superfamily, and acts as a determinative uptake pathway of metformin pharmacokinetics [50].”

6) The subtitle of the manuscript on Section 7 is confusing. I suggest the author change them to ‘Effect of metformin in other cells and biomaterials’.

Thank you for the suggestion, we have changed the subtitle of Section 7 (now Section 6 line 428).

7) There are more foreseeable future applications of metformin neglected in this review: quantifying metformin both in vivo and in vitro, how to apply metformin to its target, and its release curve, etc. This should be further addressed in the ‘clinical application’ section.

Yes, good points. We have added the following parts in Section 7 line 510:“Bone defects caused by tumor have higher recurrence rate, because it is difficult to repair, and to eliminate residual tumor cells. Metformin-loaded poly(L-lactic acid) (PLLA)/nanoscale hydroxyapatite (nHA)/metformin (MET) nanocomposite scaffold could simultaneously enhance bone repair and bone tumor inhibition [67].”

Thank you very much for your consideration.  We look forward to hearing from you.

Best Regards,

Yuxing, Bai

Reviewer 2 Report

This article entitled “Effects of metformin delivery via biomaterials on bone and dental tissue engineering ” is based on an interesting topic, well customized on the field of tissue engineering. The authors have well designed their starting hypothesis; however, I do suggest some improvements that may have impact also on the proper understanding of the main aspects of this paper.

- Currently, a growing interest has been paid on extracellular environment and signalling, specifically, the research is interested on nanoparticles and exosomes: please discuss about it  (See: Codispoti, B., Marrelli, M., Paduano, F., Tatullo, M. (2018). NANOmetric BIO-Banked MSC-Derived Exosome (NANOBIOME) as a Novel Approach to Regenerative Medicine. Journal of clinical medicine, 7(10), 357.) and how such aspect may have influence on your delivery experimental model.

- Authors have pushed their work on tissue regeneration; in this landscape, a pivotal role could be played by inflammation and its triggers. Authors should briefly report how this could affect the overall impact of their research.

- Authors may also widely and in details compare their approach with other promising smart-biomaterials, such as the Phosphorene or the Borophene. (Tatullo, M., Genovese, F., Aiello, E., Amantea, M., Makeeva, I., Zavan, B., Rengo, S., & Fortunato, L. (2019). Phosphorene Is the New Graphene in Biomedical Applications. Materials (Basel, Switzerland), 12(14), 2301.  – AND - Tatullo, M.; Zavan, B.; Genovese, F.; Codispoti, B.; Makeeva, I.; Rengo, S.; Fortunato, L.; Spagnuolo, G. Borophene Is a Promising 2D Allotropic Material for Biomedical Devices. Appl. Sci. 2019, 9, 3446. )

- Main limitations should be reported

Minor Comments:

- Please explain all the acronyms throughout the text

- Future prospects must be also related to clinical strategies: please improve this section accordingly.

Author Response

Prof. Claire Lim

Editor, International Journal of Molecular Sciences

Dear Prof. Lim:

I hope you are doing very well.  Thank you very much for a favorable review on our paper, ijms-2044947, entitled: “Effects of metformin delivery via biomaterials on bone and dental tissue engineering.” The reviewers provided excellent comments, all of which have been addressed in the revised paper.  Below is a point-by-point response to the review comments.  Each comment is followed by our response in a red color.  The revised sections in the paper are also highlighted in red.

Reviewer 2

This article entitled “Effects of metformin delivery via biomaterials on bone and dental tissue engineering ” is based on an interesting topic, well customized on the field of tissue engineering. The authors have well designed their starting hypothesis; however, I do suggest some improvements that may have impact also on the proper understanding of the main aspects of this paper.

Thank you very much for an excellent review and comments that have helped significantly improve our paper.

1) Currently, a growing interest has been paid on extracellular environment and signalling, specifically, the research is interested on nanoparticles and exosomes: please discuss about it  (See: Codispoti, B., Marrelli, M., Paduano, F., Tatullo, M. (2018). NANOmetric BIO-Banked MSC-Derived Exosome (NANOBIOME) as a Novel Approach to Regenerative Medicine. Journal of clinical medicine, 7(10), 357.) and how such aspect may have influence on your delivery experimental model.

Yes, good points. We have added the following parts in Section 6 line 439:“Furthermore, MSC-derived nanoparticles and exosomes could transport proteins, miRNA, mRNA, and exert same features as MSCs [59]. Hence, further study needs to investigate the combination of exosomes and metformin-loaded scaffold.”

2) Authors have pushed their work on tissue regeneration; in this landscape, a pivotal role could be played by inflammation and its triggers. Authors should briefly report how this could affect the overall impact of their research.

Good points. We have added the following parts in Section 4 line 343:“The inhibition of inflammation and its triggers also play important roles in bone tissue engineering. A recent study showed that metformin could also inhibit bacterial growth and downregulate inflammation to a certain extent [46].”

3) Authors may also widely and in details compare their approach with other promising smart-biomaterials, such as the Phosphorene or the Borophene. (Tatullo, M., Genovese, F., Aiello, E., Amantea, M., Makeeva, I., Zavan, B., Rengo, S., & Fortunato, L. (2019). Phosphorene Is the New Graphene in Biomedical Applications. Materials (Basel, Switzerland), 12(14), 2301.  – AND - Tatullo, M.; Zavan, B.; Genovese, F.; Codispoti, B.; Makeeva, I.; Rengo, S.; Fortunato, L.; Spagnuolo, G. Borophene Is a Promising 2D Allotropic Material for Biomedical Devices. Appl. Sci. 2019, 9, 3446. )

Yes, good point. We have added the following parts in Section 6 line 479:“Other non-cell materials, such as scaffolds, macromolecule polymers, graphene, and resins could also carry and deliver metformin in bone tissue engineering. Studies revealed that graphene, a promising smart biomaterial, had both antibacterial effects and spectacular physical peculiarities [62]. In addition, black phosphorus (BP), an allotrope of phosphorus, showed the same properties but had better biocompatibility, biodegradability, and biosafety [63]. Furthermore, borophene could also act as a prototype for metformin-loaded scaffold materials [64].

4) Main limitations should be reported.

Yes, good point. We have added the following parts in Section 4 line 337 and Section 7 line 521, respectively :“However, the exact concentration and the pharmacokinetic curve of metformin should be defined in specific clinical applications. Further studies on the effect of metformin concentration are still needed.”; “However, the concentration-dependent osteogenic effects of metformin are still unclear. Further studies are needed to use animal models to investigate how metformin interacts with various types of target tissues, and to determine the osteogenic and angiogenic effects of metformin.”

5) Please explain all the acronyms throughout the text

Thank you for a careful review. We have explained and double-checked all the acronyms. We are very grateful to Reviewer 2 for the excellent and thorough review with helpful comments. For example, we have revised “OCT” to “organic cation transporter”, and “SLC superfamily” to “solute carrier (SLC) superfamily”.

6) Future prospects must be also related to clinical strategies: please improve this section accordingly.

Good point. We have added the following parts in Section 7 line 510 and line 521, respectively:“Bone defects caused by tumor have higher recurrence rates, due to the difficulty to repair and eliminate residual tumor cells. Metformin-loaded poly (L-lactic acid) (PLLA)/nanoscale hydroxyapatite (nHA)/metformin (MET) nanocomposite scaffold could simultaneously enhance bone repair and suppress bone tumors [67].”; and “However, the concentration-dependent osteogenic effects of metformin are still unclear. Further studies are needed to use animal models to investigate how metformin interacts with various types of target tissues, and to determine the osteogenic and angiogenic effects of metformin.”

Best Regards,

Yuxing, Bai

Reviewer 3 Report

Manuscript ID ijms-2044947 by Zhu M. et al, is a review of the in vitro effects of Metformin on different stem cells implicated in dental bone tissue engineering and regeneration. The topic is very interesting, although I found the manuscript not well organized and somewhat confusing. Therefore, I suggest to deeply revise the content organization considering the following suggestions:

-Each section should be summarized, carefully avoiding any repetition. For example, in section 2, paragraphs from line 107 on p. 3 to line 130 on p. 4, should be reduced to the main concepts. The description of bone TE approaches (lines 119-131) is redundant. The phrase in line 132 "Metformin enhances osteogenesis in bone tissue engineering" is a repetition and should be removed. Besides summarizing each section, I suggest also to include section 4 main concepts into the discussion of section 2, so highlighting in that context the importance of a proper Metformin dosage. Finally, sections 5 and 6 (to be summarized) should be combined in one section.

-on p. 4 phrases from line 141 to line 148 seem to be speculative. Those outlined MF properties are only supposed or are there studies confirming them? In that case, please add relevant references. The same issue regards phrase on p. 7, lines 195-196.

Author Response

December 03, 2022

Prof. Claire Lim

Editor, International Journal of Molecular Sciences

Dear Prof. Lim:

I hope you are doing very well.  Thank you very much for a favorable review on our paper, ijms-2044947, entitled: “Effects of metformin delivery via biomaterials on bone and dental tissue engineering.” The reviewers provided excellent comments, all of which have been addressed in the revised paper.  Below is a point-by-point response to the review comments.  Each comment is followed by our response in a red color.  The revised sections in the paper are also highlighted in red.

Reviewer 3

Manuscript ID ijms-2044947 by Zhu M. et al, is a review of the in vitro effects of Metformin on different stem cells implicated in dental bone tissue engineering and regeneration. The topic is very interesting, although I found the manuscript not well organized and somewhat confusing. Therefore, I suggest to deeply revise the content organization considering the following suggestions:

Thank you very much for an excellent review and comments that have helped significantly improve our paper.

1) Each section should be summarized, carefully avoiding any repetition. For example, in section 2, paragraphs from line 107 on p. 3 to line 130 on p. 4, should be reduced to the main concepts. The description of bone TE approaches (lines 119-131) is redundant. The phrase in line 132 "Metformin enhances osteogenesis in bone tissue engineering" is a repetition and should be removed.

Yes, good points. We have reduced line 107-132 to the main concepts, deleted the repetition and the redundant parts, revised as the following parts in Section 2 line 107-:“Bone defects are often caused by trauma, tumor, and other chronic diseases, such as periodontitis. Periodontitis is an inflammatory disease caused by the plaque [25]. Traditional periodontal treatments have the ability to alleviate inflammation and suppress the progression of inflammatory process to a certain extent [26]. However, when bone loss caused by periodontitis reaches a critical size, large bone defects cannot be repaired if treated only by the traditional treatments mentioned above. It remains an unsolved problem in restoring the structure and function of periodontitis bone loss.”

2) Besides summarizing each section, I suggest also to include section 4 main concepts into the discussion of section 2, so highlighting in that context the importance of a proper Metformin dosage.

Good points. We have added the following parts in Section 2 line 182:“The optimal concentration of metformin in bone repair and regeneration remains unknown. There are distinctive differences regarding its dosage either for different target tissues, or different administration methods. Notably, an oral dose of 500-1500 mg metformin was widely used in clinical practice [29]. The following section reviewed both in vitro doses for local administration and in vivo doses of metformin.”

3) Finally, sections 5 and 6 (to be summarized) should be combined in one section.

Yes, good points. We have deleted the subtitle of Section 6, and revised the subtitle of Section 5 into “Effect of metformin on dental-derived stem cells”, deleted several redundant paragraphs, and summarized section 5.

4) On p. 4 phrases from line 141 to line 148 seem to be speculative. Those outlined MF properties are only supposed or are there studies confirming them? In that case, please add relevant references. The same issue regards phrase on p. 7, lines 195-196.

We are sorry for the mistakes. We added supporting references (reference No.27 and No.9) regarding MF properties from line 141 to 148.

And deleted the phrase on p. 7, lines 195-196.

Thank you very much for your consideration.  We look forward to hearing from you.

Best Regards,

Yuxing, Bai

Reviewer 4 Report

The aim of this paper is make a review article on the effects of metformin on stem cells and bone tissue engineering. They analyzed effects of Metformin on bone repair and regeneration, and on periodontal ligament and dental-derived stem cells. They also analyzed effects of different metformin concentrations and clinical application of metformin in bone and dental tissue regeneration.

The review is clear, comprehensive and current.

References cited are recent pubblications. The reference number 5 is omitted

Figures, tables are appropiate and easy to interpret and understand.

The English language is appropriate and understandable.

The statements and conclusions are drawn coherent and supported by the listed citations

The conclusions are interesting for the readership of the journal.

The authors offer interesting suggestions for continuing research on the alternative use of metfformine in bone and dental tissue regeneration.

Author Response

December 03, 2022

Prof. Claire Lim

Editor, International Journal of Molecular Sciences

Dear Prof. Lim:

I hope you are doing very well.  Thank you very much for a favorable review on our paper, ijms-2044947, entitled: “Effects of metformin delivery via biomaterials on bone and dental tissue engineering.” The reviewers provided excellent comments, all of which have been addressed in the revised paper.  Below is a point-by-point response to the review comments.  Each comment is followed by our response in a red color.  The revised sections in the paper are also highlighted in red.

Reviewer 4

The aim of this paper is making a review article on the effects of metformin on stem cells and bone tissue engineering. They analyzed effects of Metformin on bone repair and regeneration, and on periodontal ligament and dental-derived stem cells. They also analyzed effects of different metformin concentrations and clinical application of metformin in bone and dental tissue regeneration. The review is clear, comprehensive and current.

References cited are recent publications. The reference number 5 is omitted

Figures, tables are appropriate and easy to interpret and understand.

The English language is appropriate and understandable.

The statements and conclusions are drawn coherent and supported by the listed citations

 The conclusions are interesting for the readership of the journal.

The authors offer interesting suggestions for continuing research on the alternative use of metformin in bone and dental tissue regeneration.

Thank you very much for an excellent review and comments that have helped significantly improve our paper.

1) The reference number 5 is omitted

We are sorry for the mistake, we have corrected and double checked all references. We are grateful to Reviewer 4 for the excellent review with helpful comments.

Thank you very much for your consideration.  We look forward to hearing from you.

Best Regards,

Yuxing, Bai

Round 2

Reviewer 2 Report

NO issues

Author Response

December 07, 2022

Prof. Claire Lim

Editor, International Journal of Molecular Sciences

Dear Prof. Lim:

I hope you are doing very well.  Thank you again for favorable reviews on our paper, ijms-2044947, entitled: “Effects of metformin delivery via biomaterials on bone and dental tissue engineering.” The reviewers provided excellent comments, all of which have been addressed in the revised paper.  Below is a point-by-point response to the review comments.  Each comment is followed by our response in a red color.  The revised sections in the paper are also highlighted in red.

Reviewer 2

NO issues.

Thank you again for the excellent review and comments that have helped significantly improve our paper.

Thank you very much for your consideration.  We look forward to hearing from you.

Best Regards,

Yuxing, Bai

Reviewer 3 Report

Authors have modified the manuscript according to my suggestions. I have noticed that Authors decided to keep section 4 roughly unmodified (my suggestion was to include it in section 2), but in my opinion it is too long and should be somewhat reduced to the main concepts to be readily understood. A table with the main metformin studied doses and effects could be of help for the readers, and also for the authors to summarize the section contents

Author Response

December 07, 2022

Prof. Claire Lim

Editor, International Journal of Molecular Sciences

Dear Prof. Lim:

I hope you are doing very well.  Thank you again for favorable reviews on our paper, ijms-2044947, entitled: “Effects of metformin delivery via biomaterials on bone and dental tissue engineering.” The reviewers provided excellent comments, all of which have been addressed in the revised paper.  Below is a point-by-point response to the review comments.  Each comment is followed by our response in a red color.  The revised sections in the paper are also highlighted in red.

Reviewer 3

Authors have modified the manuscript according to my suggestions.

Thank you again for the excellent review and comments that have helped significantly improve our paper.

1) I have noticed that Authors decided to keep section 4 roughly unmodified (my suggestion was to include it in section 2), but in my opinion it is too long and should be somewhat reduced to the main concepts to be readily understood.

Yes, good points. Section 4 is too long, we followed your advice, reduced to the main concepts, and included it into Section 2 line 163:“Clinically relevant doses of metformin were demonstrated to be associated with the osteogenic differentiation and mineralization of iPSC‐MSCs [30]. Notably, there are large differences in the optimal concentration of metformin for different cell types, suggesting that it is important to explore the most suitable concentration of metformin for its application in tissue engineering [30]. It was reported that metformin promotes osteoblastic differentiation through AMPK signaling at doses ranging from 0.5 to 500 μM [31]. This was specifically demonstrated via a dose-dependent effect on cell proliferation, as well as an increase in extracellular mineral nodule formation and the most-recognized osteogenic markers. Our previous study found that after treating MSCs cells with increasing doses of metformin (0–20 μM), metformin increased cell viability in a dose-dependent manner [31]. These findings underscore the importance of using therapeutically relevant doses when attempting to extrapolate in vitro results to the human clinical setting. Pharmacokinetic studies verified that within 2–4 h after an oral dose of 500–1500 mg, plasma concentrations of metformin in patients ranged from 2.7 to 20 μM [32].

Furthermore, sustained cell growth was observed under different treatment conditions. Sun et al. [7] 125 μM was the optimal concentration of metformin for osteogenic differentiation and could promote implant osseointegration in the rats. At concentrations over 200 µM, metformin inhibited the osteogenic differentiation of BMSCs [7]. The pro-osteogenic function of metformin during in vitro and in vivo osteogenesis of adipose-derived stromal cells was assessed [33]. The in vitro experiments showed that metformin added into the culture medium at a concentration of 500 µM promoted the differentiation of adipose-derived stromal cells into bone-forming cells and increased the formation of mineralized extracellular matrix. The in vivo models revealed that metformin at a dose of 250 mg/kg/day accelerated bone healing and facilitated new bone callus formation at fracture sites in a rat cranial defect model [43]. It was also reported that 100 µM metformin had a prominent positive effect on the osteogenic differentiation, and a negative effect on the adipogenic differentiation, of PDLSCs, which has important implications for the application of metformin in PDLSC-based osteogenesis and bone regeneration [34] (Table 1).

Local administration of metformin has also been studied. A novel guided bone regeneration (GBR) membrane containing polycaprolactone (PCL) and polyvinyl alcohol (PVA) with different concentrations of metformin was developed to improve osteogenic capability [13]. The results of that study confirmed the high potential of co-cultured stem cells with 10 wt% metformin PCL/PVA membranes for GBR applications [13]. Another model system containing 20 wt% metformin, incorporated into resin, was established for localized delivery of drugs into bone defect sites. The metformin-loaded resin could help restore the tooth cavity, provide protection for dental pulp, and prevent microleakage during restoration [35]. The effects of topical application of metformin on promoting new bone formation and remodeling were also evaluated [36]. Tendon-bone interface healing involves fine osteogenesis at the repair site. In vitro experiments used BMSCs with different concentrations of metformin (0, 10, 50, 100, and 200 μM) cultured together. The results showed that the mechanical strength and new bone volume of the metformin-applied group were significantly higher than those of the control group. However, the exact concentration and the pharmacokinetic curve of metformin should be defined in specific clinical applications. Further studies on the effect of metformin concentration are still needed.”

2) A table with the main metformin studied doses and effects could be of help for the readers, and also for the authors to summarize the section contents

Good points. We have added table 1:

Thank you very much for your consideration.  We look forward to hearing from you.

Best Regards,

Yuxing, Bai
